# PD-L1 Expression in Non-Small Cell Lung Cancer: Data from a Referral Center in Spain

**DOI:** 10.3390/diagnostics11081452

**Published:** 2021-08-11

**Authors:** Karmele Saez de Gordoa, Ingrid Lopez, Marta Marginet, Berta Coloma, Gerard Frigola, Naiara Vega, Daniel Martinez, Cristina Teixido

**Affiliations:** 1Thoracic Oncology Unit, Department of Pathology, Hospital Clínic of Barcelona, 08036 Barcelona, Spain; saezdegord@clinic.cat (K.S.d.G.); ilopezs@clinic.cat (I.L.); mmarginet@clinic.cat (M.M.); bcoloma@clinic.cat (B.C.); frigola@clinic.cat (G.F.); nvega@clinic.cat (N.V.); dmartin1@clinic.cat (D.M.); 2Translational Genomic and Targeted Therapeutics in Solid Tumors, August Pi i Sunyer Biomedical Research Institute (IDIBAPS), 08036 Barcelona, Spain

**Keywords:** PD-L1, immunotherapy, immunohistochemistry, lung cancer, NSCLC

## Abstract

Anti-programmed cell death (PD1)/ligand-1 (PD-L1) checkpoint inhibitors have improved the survival of non-small cell lung cancer (NSCLC) patients. Additionally, PD-L1 has emerged as a predictive biomarker of response. Our goal was to examine the histological features of all PD-L1 cases of NSCLC analyzed in our center between 2017 and 2020, as well as to correlate the expression values of the same patient in different tested samples. PD-L1 immunohistochemistry (IHC) was carried out on 1279 external and internal samples: 482 negative (tumor proportion score, TPS < 1%; 37.7%), 444 low-expression (TPS 1–49%; 34.7%) and 353 high-expression (TPS ≥ 50%; 27.6%). Similar results were observed with samples from our institution (N = 816). Significant differences were observed with respect to tumor histological type (*p* = 0.004); squamous carcinoma was positive in a higher proportion of cases than other histological types. There were also differences between PD-L1 expression and the type of sample analyzed (surgical, biopsy, cytology; *p* < 0.001), with a higher frequency of negative cytology. In addition, there were cases with more than one PD-L1 determination, showing heterogeneity. Our results show strong correlation with the literature data and reveal heterogeneity between tumors and samples from the same patient, which could affect eligibility for treatment with immunotherapy.

## 1. Introduction

Recent advances in cancer immunotherapy with anti-programmed cell death (PD1) and anti-programmed cell death ligand 1 (PD-L1) inhibitors have shown significant survival benefits in advanced non-small cell lung carcinoma (NSCLC), leading to impressive outcomes [1,2,3]. As a result, immunotherapy has become part of the clinical approach for management of NSCLC, either in monotherapy or in combination therapies [4,5,6].

In order to determine patients’ suitability for immunotherapy, avoid unnecessary exposure to possible toxicities and provide maximum effectiveness, different biomarkers have been suggested for NSCLC [7]. Factors that affect choice of treatment in NSCLCs that lack an oncogene-addicted alteration include PD-L1 status, stage of the disease and histology type.

Testing for PD-L1 expression is the current standard for identifying individuals with advanced NSCLC who are more likely to respond to immunotherapy (among both treatment-naive patients and subsequent-line treatment settings) [8]. Pembrolizumab was the first immune checkpoint inhibitor (ICI) approved as a single agent in the first-line setting in advanced NSCLC patients with no epidermal growth factor receptor (*EGFR*) or anaplastic lymphoma kinase (*ALK*) genomic tumor aberrations and high expression levels of PD-L1 (tumor proportion score, TPS ≥ 50%) [9,10]—which accounts for almost one-third of advanced NSCLC patients. Furthermore, recent first-line studies of pembrolizumab, in combination with platinum doublets in metastatic non-squamous and squamous NSCLC, benefited from the addition of ICIs to chemotherapy irrespective of PD-L1 expression [11,12,13,14].

Atezolizumab, cemiplimab and nivolumab are also considered for first-line treatment in advanced NSCLC without *EGFR* or *ALK* genomic alterations. Treatments include: atezolizumab with bevacizumab and chemotherapy in non-squamous NSCLC [15,16], atezolizumab with carboplatin and nab-paclitaxel chemotherapy [17], atezolizumab in monotherapy if PD-L1 TPS ≥ 50% or PD-L1-stained tumor-infiltrating immune cells cover ≥10 percent of the tumor area [6,18] and cemiplimab monotherapy if PD-L1 TPS ≥ 50% [19]. Nivolumab plus ipilimumab is another alternative for advanced NSCLC with PD-L1 TPS ≥ 1%; however, in the CheckMate-227 trial, this combination demonstrated a superior survival rate compared with chemotherapy, irrespective of PD-L1 expression [20]. On the other hand, pembrolizumab, atezolizumab and nivolumab are approved in the second-line setting, but only pembrolizumab requires PD-L1 expression in tumor cells (TPS ≥ 1%) for eligibility [21,22,23].

At the present time, the only clinically validated first-line biomarker for determining eligibility for PD1/PD-L1 ICIs for patients with advanced NSCLC is PD-L1 expression by IHC on formalin-fixed, paraffin-embedded (FFPE) sections. However, the use of PD-L1 as a biomarker is not exempt from hurdles; different testing systems have been used and different scoring and cut-off determinations have been suggested. Additionally, tumor heterogeneity in PD-L1 expression presents a challenge [24,25].

In the present study, we examined the PD-L1 expression rates of NSCLC cases conducted at our center between 2017 and 2020. Our goal was to study whether the population that could benefit from targeted treatment was in concordance with published results and to provide the opportunity to obtain information on the possible relationships of PD-L1 expression with tumor histology, the type of sample analyzed and the site of the sample. In addition, we studied the correlation between the expression values of the same patient when different samples were tested for PD-L1 IHC.

## 2. Materials and Methods

### 2.1. Patients and Samples

Patients with a confirmed diagnosis of NSCLC who underwent a PD-L1 expression test at Hospital Clinic Barcelona (Barcelona, Spain) from January 2017 to July 2020 were included in the study. Cell blocks or FFPE tissue samples were analyzed regardless of sample site and type. Cell blocks were prepared using HistoGel with normal saline, and cells were obtained by fine-needle aspiration, pleural and pericardial effusions or bronchoalveolar lavages.

Clinical and pathological data were obtained from the electronic medical records of our institution. Extracted data were patient age, gender, type of sample (biopsy, cytology or surgical specimen), site of the sample (primary tumor in the lung, lymph node metastasis or distant metastasis) and histological type (adenocarcinoma, squamous cell carcinoma or other NSCLC—which includes NSCLC not otherwise specified (NOS), pleomorphic carcinoma, large cell carcinoma and other subtypes of NSCLC).

The study was conducted in accordance with the principles of the Declaration of Helsinki under approval from the Internal Review Board of the Hospital Clinic Barcelona (HCB/2016/0354; approved on 5 November 2016).

### 2.2. PD-L1 Immunohistochemistry

Immunohistochemistry (IHC) was performed on 4 µm sections, using the in vitro diagnostic (IVD) Dako PD-L1 IHC 22C3 pharmDx assay or Dako 28-8 pharmDx (Agilent Technologies Dako, Glostrup, Denmark), performed on Dako Autostainer Link 48. Patients with PD-L1 testing on cytology were not required to have confirmation on biopsy when available. Human tonsil tissue was used as an external control.

The histological evaluation was performed by pathology specialists trained in PD-L1 evaluation. The cytological samples were evaluated by a cytotechnician and a cytopathologist, both experts in lung cancer cytology. PD-L1 expression in tumor cells was evaluated according to the guidelines [25,26], and the report included the proportion of tumor cells with partial or complete membrane staining (TPS—tumor proportion score) [27]. This staining was then classified into three groups according to the TPS: negative (0 and <1); low expression (1–49); and high expression (≥50). Cases with less than 100 viable tumor cells were regarded non-assessable for PD-L1 expression.

### 2.3. Statistical Analysis

Descriptive statistics, including mean and median for continuous variables, or percentages and frequencies for categorical variables, were tabulated and are presented. PD-L1 result was analyzed as a three-category variable, a two-category variable (for TPS 1% and TPS 50% cut-offs) and as a continuous variable. When analyzed categorically, Chi-square test or Fisher’s exact test were performed for comparing group frequencies. When analyzed as a continuous variable, Wilcoxon and Kruskal–Wallis tests were performed, the first for two-group comparison and the second for more-than-two-group comparison. Dunn’s test was performed in the case of significant comparisons with Kruskal–Wallis test. In cases with more than one measurement, Pearson’s correlation coefficient was calculated to evaluate concordance between continuous variables and Cohen’s kappa coefficient of agreement for categorical results, with the following levels of concordance: poor (k = 0.00), slight (k = 0.00–0.20), fair (k = 0.21–0.40), moderate (k = 0.41–0.60), substantial (k = 0.61–0.80) and almost perfect (k = 0.81–1.00). For the statistical analysis, the R program (4.0.3 version; R Foundation for Statistical Computing, Vienna, Austria) was used. All tests were performed two-sided, at a significance level of 0.05 and calculated at confidence level 1 − α = 0.95.

## 3. Results

### 3.1. Patient Cohort

A total of 1307 requests for PD-L1 IHC were received at our center from January 2017 to July 2020, corresponding to 1153 patients with advanced NSCLC. Twenty-eight samples were non-assessable because they had less than 100 viable tumor cells (2.14%; Figure 1).

Of the 1279 PD-L1 evaluable samples, 463 were from external laboratories and there was no access to all clinicopathological data. Among the 816 samples analyzed in our center, throughout the study period, an increasing number of PD-L1 test requests was observed: 2017, N = 211; 2018, N = 240; 2019, N = 262; and during the first half of 2020, N = 103.

### 3.2. PD-L1 Expression and Patient Characteristics

PD-L1 staining was observed in both tumor cells and immune cells, but only tumor cells were assessed. In the 1279 NSCLC samples tested at our institution, the proportion of negative results was 37.7% (N = 482); low expression 34.7% (N = 444); and high expression 27.6% (N = 353). Samples tested were biopsies (N = 543; 42.5%), surgical specimens (N = 411; 32.1%) and cell blocks (N = 325; 25.4%).

Significant differences were found when comparing PD-L1 expression with the type of sample when analyzed as a three-category analysis (*p* < 0.001). A higher PD-L1 positivity (TPS ≥ 1%) was identified in biopsies (N = 372; 68.5%) than surgical specimens (N = 261; 63.5%) and cytologies (N = 164; 50.5%), *p* < 0.001. Additionally, we also observed differences when analyzing the PD-L1 TPS 50% cut-off point, *p* < 0.001: biopsies (N = 186), 34.3% vs. surgical specimens (N = 97), 23.6% vs. cytologies (N = 70), 21.5%, and as a continuous variable (*p* < 0.001).

We then performed the analysis including only the patients from our hospital (N = 816). As mentioned above, only complete clinicopathological data were obtained from the samples acquired in our center, Table 1. Most of the patients were men (69.7%) and older than 65 years (56.7%). The histology was adenocarcinoma in 58.2% of the patients and the majority of the types of sample analyzed were biopsies (39.22%).

In our PD-L1 analysis, 323 cases (39.6%) were negative (TPS < 1%); 270 (33.1%) showed low expression (TPS 1–49%); and 223 (27.3%) showed high expression (TPS ≥ 50%).

No differences were found between PD-L1 staining and the gender or the age of the patients. Statistically significant differences were observed in terms of the different type of samples used for PD-L1 staining (*p* < 0.001), with cytology resulting in a negative result more frequently (Figure 2a). Differences were also observed between PD-L1 expression and histological type (*p* = 0.002). The frequency of a negative result was higher in adenocarcinomas, while cases with NOS or mixed NSCLC histologies had the highest expression results (TPS ≥ 50%; Figure 2c). On the contrary, we did not find variations in different sites of sample (*p* = 0.458).

The same differences in the frequencies of PD-L1 staining were observed when we analyzed it as a dichotomic variable, using the thresholds of TPS 1% and TPS 50% (Figure 2a,c), and as a continuous variable (Figure 2b,d). Regarding the histological type of the tumor, squamous cell carcinoma and other NSCLC presented higher PD-L1 TPS than adenocarcinoma, which was statistically significant. These results are presented in Table 2.

### 3.3. Agreement between the PD-L1 Results in Cases with More Than One Measurement in Different Samples

Among the cases included in our study, 191 samples corresponded to 88 unique patients. The samples were paired according to the type of sample (34 cases of biopsy and cytology, 11 cases of biopsy and surgical resection and 19 cases of cytology and surgical specimen), and site of the sample (six cases of lymph node metastases and distant metastases, 21 cases of lymph node metastases and primary tumor and 17 cases of primary tumor and distant metastases), resulting in 108 pairs of samples for comparison. Figure 3 shows representative IHC images of these comparative groups.

Most of the pairs of samples analyzed were between different types of samples (N = 64; 59.3%). The three comparisons made between the types of sample had a good Pearson correlation coefficient (*p*-value < 0.05), which measured PD-L1 TPS as a continuous variable (Figure 4a–c). However, the agreement calculated with Cohen’s kappa coefficient for the TPS 1% and the TPS 50% cut-off points was much lower. Cytology and surgical specimen comparison had a moderate agreement for the PD-L1 TPS 1% cut-off point (k = 0.451) and a substantial agreement for the PD-L1 TPS 50% cut-off (k = 0.617). Comparisons of biopsy and cytology had moderate agreement (k = 0.436 for TPS 1% cut-off and k = 0.544 for TPS 50% cut-off). Biopsy and surgical specimen had slight agreement for TPS 1% cut-off (k = 0.1) and moderate agreement for TPS 50% cut-off (k = 0.421).

Forty-four pairs of samples were matched according to site of sample (40.7%). The primary lung tumor compared to the lymph node (Figure 4d) had a significant result with Pearson’s correlation coefficient (*p*-value < 0.001), but had a moderate agreement for PD-L1 TPS 1% and TPS 50% cut-offs (k = 0.512 and k = 0.576, respectively). Lung and distant metastases groups (Figure 4e) had a significant result with Pearson’s correlation coefficient (*p*-value < 0.001), but had a moderate agreement for TPS 1% cut-off (k = 0.433) and a substantial agreement for TPS 50% cut-off (k = 0.643). The PD-L1 status comparison between samples from lymph node metastases and distant metastases (Figure 4f) had the lowest number of cases (N = 6) and did not show significant results with Pearson’s correlation coefficient (*p* = 0.718), and Cohen’s kappa agreement was poor or fair (k = 0 for PD-L1 TPS 1% cut-off and k = 0.286 for PD-L1 TPS 50% cut-off).

## 4. Discussion

PD-L1 expression by IHC provides a reproducible result that, in many settings, is applied to identify patients eligible for treatment with ICIs [7]. Although trials have demonstrated that PD-L1 expression is associated with a higher likelihood of response to ICIs, it does not guarantee response in those with high PD-L1 expression in the tumor, nor does it eliminate the possibility of response in those tumors that lack PD-L1 expression.

Due to the importance of PD-L1 as a crucial protein for tumor immune escape, and the fact that its presence indicates a potential target for ICIs in NSCLC, in the present retrospective study we collected all the PD-L1 tests performed as a daily routine diagnosis in our institution with NSCLC, and the data obtained were analyzed. A total of 1279 cases was assessed and we observed that each year the number of PD-L1 tests increased. The proportion of PD-L1 expression observed in our cohort is similar to other published studies [28,29], with a PD-L1 negative result in 39–41% of cases, low expression in 30–38% of cases and high expression in 21–30% of cases. Our results suggest that PD-L1 staining pattern varies according to tumor histology, with higher expression in squamous cell carcinomas and other types of NSCLC than adenocarcinomas. This observation was also observed and published by other authors. Tsao et al. [30] found a higher expression of PD-L1 in poorly differentiated early-stage NSCLC samples, including large cell, adenosquamous, pleomorphic and sarcomatoid carcinomas. The study by Hong et al. [29] observed a higher expression of PD-L1 in squamous cell carcinomas than adenocarcinomas, while the study recently published by Zheng and colleagues [31] found a higher positivity of PD-L1 in squamous cell carcinomas and other histological subtypes of NSCLC compared to adenocarcinomas. Nonetheless, these results seem controversial, as other studies have not found these differences [32], or have reported higher staining in adenocarcinomas [33].

Most of our patients were male and no differences were observed between the genders, while one study, which had a majority of female patients, reported differences in PD-L1 positivity among both genders [34]. On the contrary, we found statistically significant differences between the type of sample and PD-L1 TPS, as has been reported by other authors [29,32]. Many NSCLC patients are diagnosed only with a cytology sample, so these samples are widely used for molecular testing, including PD-L1 testing, with high assessment rates [33,35]. In fact, in our cohort, cytological samples represent more than 25% of cases. According to our results, cytological samples tend to show less PD-L1 staining in tumor cells, as several authors have previously observed [27,36]. The heterogeneity of PD-L1 expression is a recognized diagnostic challenge, but this finding raises concerns that cytology samples may not be processed well at our institution; it may be that the procedure harms the tumor cell membrane and cells do not have a completed cell membrane, or the samples from our study have not provided sufficient tumor cells for accurate PD-L1 evaluation, to counteract the tumor heterogeneity. Despite this, because of the concern for undertreatment and the fact that PD-L1 TPS interpretation in cytological samples is challenging, these cases were thoroughly examined and evaluated by experts in lung cancer cytology.

Contradictory findings have been reported regarding PD-L1 expression in biopsy versus surgical specimens. Although Gradecki et al. [37] have reported a concordance of PD-L1 expression between core biopsy and resection specimens, other authors have found discrepancies between different tissue sampling methods. Gagne et al. [32] found a greater PD-L1 expression in cell blocks compared to biopsies and surgical specimens, but supporting our findings, a higher proportion of PD-L1 TPS ≥ 50% was found in biopsies compared with surgical specimens. Along these lines, Zheng et al. [31] found that PD-L1 expression was higher in biopsies than in surgically resected specimens in a real-world study in China. The discrepancy found between PD-L1 expression in biopsy compared to resection specimens could be due to intratumoral heterogeneity, an advanced stage of the biopsy samples compared to the surgical specimens or the age of the samples. Older versus recent specimens may lead to an underestimation of PD-L1 status [26]. Although PD-L1 testing is recommended for samples less than three years old, a diminution in PD-L1 staining was already observed by Giunchi et al. with 1-year-old specimens [38].

On the other hand, we compared paired samples with PD-L1 results and observed a significant concordance in Pearson’s correlation coefficient with PD-L1 as a continuous variable. Cohen’s kappa coefficient reflected various degrees of agreement, mostly moderate, when analyzing PD-L1 TPS 1% and 50% cut-off points. Similar results were also found in a study by Cho et al. with significant correlation of PD-L1 expression as a continuous variable, while the concordance rate as a categorical variable was only 57% [39]. We revised our cases, and found many where a slight change in TPS value led to a different PD-L1 category. As the expression of PD-L1 is a continuous variable (not rounded, an exact number in between 0 and 100 is given), any scoring around any of the relevant thresholds will inevitably be subject to some interobserver variability. This can be a problematic issue, as treatment decisions are made according to the different cut-off points. In our center, the histological evaluation was carried out by pathology specialists trained in PD-L1 evaluation, and cases with low PD-L1 positivity (TPS < 5%), mainly in cytological samples, were thoroughly examined and another expert pathologist was consulted. It should be noted that in this scenario, in samples with a TPS < 5%, it is difficult to differentiate the positive tumor cells isolated from the immune cells that infiltrate the tumor.

Furthermore, a Pearson correlation coefficient was observed in paired cases with samples from different sites, primary lung tumor versus lymph node metastasis and primary lung tumor versus distant metastasis. Similarly, Kim et al. reported a concordance rate of 75.2% comparing primary tumor and metastases, with a moderate agreement in Cohen’s k coefficient (k = 0.433) [40], while another study from Cho et al. found no differences [39].

On the other side, Mansfield et al. found different expression patterns of PD-L1 comparing primary tumor of the lung and brain metastases [41].

In summary, the present study provides multicenter real-world data to assess the feasibility of PD-L1 testing in different histological subtypes of NSCLC samples, using different sample acquisition modes. Its main two advantages are that PD-L1 expression was evaluated by a referral center, avoiding interobserver variability, and an approved PD-L1 IVD assay was used in all tests performed, most of which were tested with the companion diagnostic (CDx) assay 22C3. One of the limitations of our study derives from the fact that concordance between the PD-L1 test in EBUS, lymph nodes and surgical samples was carried out, but was not planned to be performed in the study. This would have required more invasive sampling in patients, with a complete molecular characterization, which is currently not our standard practice. Accordingly, we have a small number of patients with more than one biopsy tested with the PD-L1 IHC technique. Another limitation is that the clinical outcomes of the patients and the mutational status of the most relevant biomarkers in advanced NSCLC were not collected, so we are not able to know whether these patients were treated with immunotherapy, or their responses. This information would be especially relevant in cases with more than one sample with contradictory results, since it would affect the patient’s eligibility for ICI therapy.

## Figures and Tables

**Figure 1 diagnostics-11-01452-f001:**
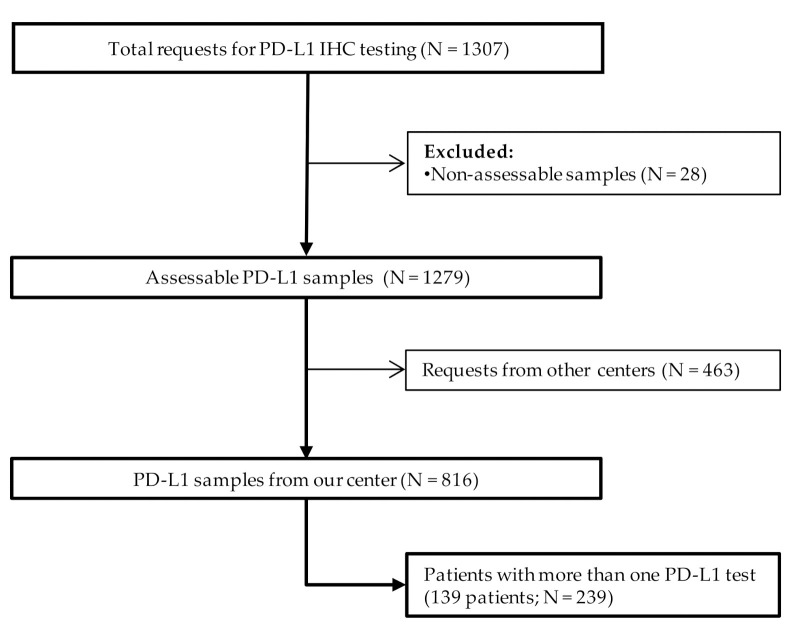
Flow chart of the patients included in the study cohort. Abbreviations: PD-L1, programmed cell death ligand 1; IHC, immunohistochemistry; NSCLC, non-small cell lung cancer.

**Figure 2 diagnostics-11-01452-f002:**
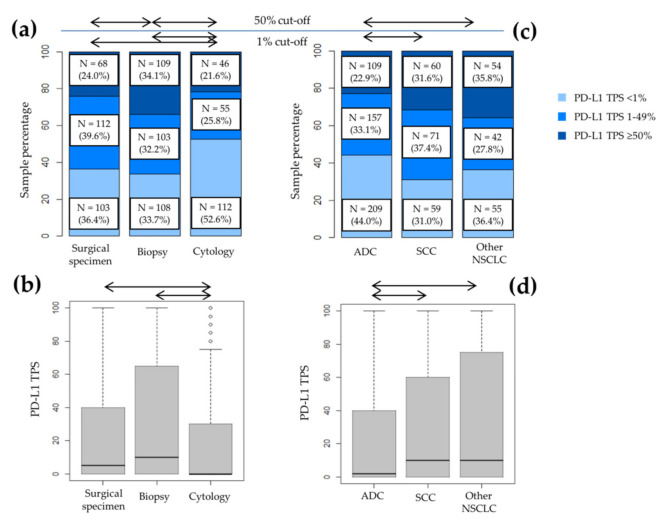
PD-L1 expression was different in each type of sample and tumor histology. (**a**) Bar plot and (**b**) box plot of the comparison of PD-L1 expression with the type of sample. The type of sample was statistically significantly related to PD-L1 expression as a three-category analysis (*p* < 0.001), separately analyzing the cut-off points TPS 1% and TPS 50% (*p* < 0.001 and *p* = 0.002, respectively), and as a continuous variable (*p* < 0.001). (**c**) Bar plot and (**d**) box plot of the relationship between PD-L1 expression and tumor histology. Differences were observed when evaluating PD-L1 as a categorical variable (both as a three-category variable, *p* = 0.002; and a two-category analysis in the cut-off points TPS 1% and TPS 50%, *p* = 0.006 and *p* = 0.003, respectively), and as a continuous variable (*p* = 0.001). Arrows indicate a statistically significant result (*p* < 0.05). Abbreviations: ADC, adenocarcinoma; SCC, squamous cell carcinoma; NSCLC, non-small cell lung cancer; PD-L1, programmed cell death ligand 1; TPS, tumor proportion score.

**Figure 3 diagnostics-11-01452-f003:**
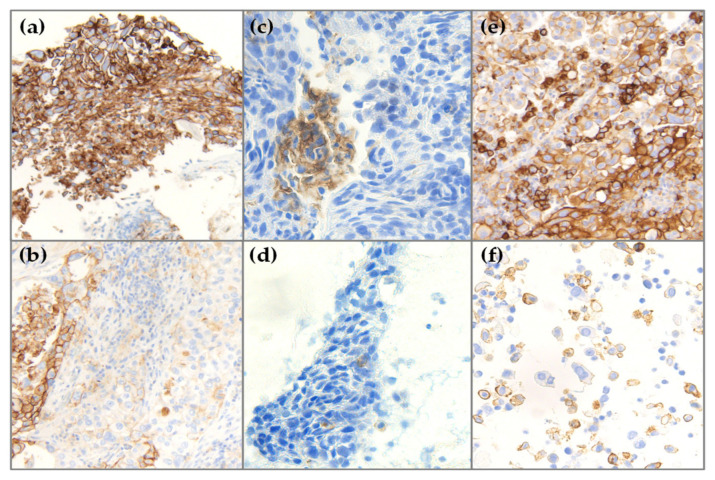
Representative images of three cases with more than one sample tested for PD-L1 expression. Case 1, (**a**) endobronchial biopsy sample of a primary lung adenocarcinoma with PD-L1 TPS 100% (10× magnification) and (**b**) surgical specimen with PD-L1 TPS 40% (10× magnification). Case 2, (**c**) endobronchial biopsy of a squamous cell carcinoma with PD-L1 TPS 5% (20× magnification) and (**d**) cytological block from a lymph node metastasis with PD-L1 TPS of <1% (20× magnification). Case 3, (**e**) surgical specimen of a pleomorphic carcinoma showing PD-L1 TPS 90% (10× magnification) and (**f**) paired cytological block from a carcinoma in the lung with PD-L1 TPS 70% (20× magnification). Abbreviations: PD-L1, programmed cell death ligand 1; TPS, tumor proportion score.

**Figure 4 diagnostics-11-01452-f004:**
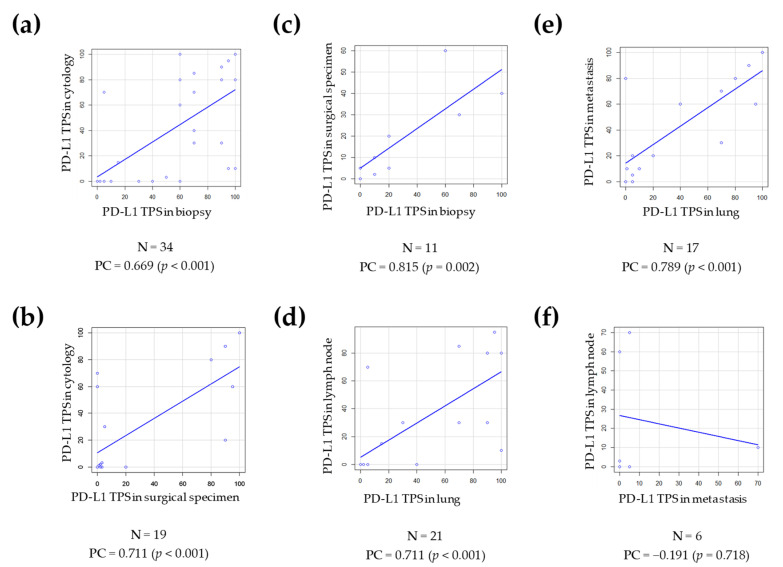
Scatter plots comparing PD-L1 TPS of paired cases with the respective Pearson’s correlation coefficient. Cases with different type of sample: (**a**) cytology and biopsy; (**b**) cytology and surgical specimen; and (**c**) surgical specimen and biopsy. Comparison of samples from different sites: (**d**) primary lung tumor and lymph node metastasis; (**e**) primary lung tumor and distant metastasis; and (**f**) lymph node and distant metastasis. Abbreviations: N, number of cases; PC, Pearson’s correlation coefficient; PD-L1, programmed cell death ligand 1; TPS, tumor proportion score.

**Table 1 diagnostics-11-01452-t001:** Patient characteristics and group comparisons of PD-L1 result (N = 816).

Clinical Characteristic		N (%)	PD-L1 TPS			*p*-Value
			<1% (%)	1–49% (%)	≥50% (%)	
Gender	Men	569 (69.73)	233 (40.9)	182 (32.0)	154 (27.1)	0.444
	Women	247 (30.27)	90 (36.5)	883 (35.6)	69 (27.9)	
Age	≤65 years	353 (43.26)	145 (41.1)	114 (32.3)	94 (26.6)	0.748
	>65 years	463 (56.74)	178 (38.4)	156 (33.7)	129 (27.9)	
Sample type	Surgical specimen	283 (34.68)	103 (36.4)	112 (39.6)	68 (24.0)	<0.001
	Biopsy	320 (39.22)	108 (33.7)	103 (32.2)	109 (34.1)	
	Cytology	213 (26.10)	112 (52.6)	55 (25.8)	46 (21.6)	
Site of sample	Lung—bronchus	541 (66.30)	207 (38.3)	180 (33.3)	154 (28.5)	0.458
	Lymph node	115 (14.09)	54 (47.0)	33 (28.7)	28 (24.3)	
	Distant metastasis	160 (19.61)	62 (38.8)	57 (35.6)	41 (25.6)	
Histology	Adenocarcinoma	475 (58.21)	209 (44.0)	157 (33.1)	109 (22.9)	0.002
	Squamous carcinoma	190 (23.28)	59 (31.1)	71 (37.4)	60 (31.6)	
	Other NSCLC	151 (18.51)	55 (36.4)	42 (27.8)	54 (35.8)	

N, number; NSCLC, non-small cell lung cancer; TPS, tumor proportion score; PD-L1, programmed cell death ligand 1.

**Table 2 diagnostics-11-01452-t002:** PD-L1 result of samples acquired at our institution (N = 816) with different types of measurements: as a continuous variable, and as a dichotomous variable (both with cut-off points TPS 1% and TPS 50%).

Clinical Characteristic		As a Continuous Variable			PD-L1 TPS 1% Cut-Off			PD-L1 TPS 50% Cut-Off		
		Median	Mean	*p*-Value	<1 N (%)	≥1 N (%)	*p*-Value	<50 N (%)	≥50 N (%)	*p*-Value
Gender	Male	3	25.17	0.157	233 (40.9)	336 (59.1)	0.226	415 (72.9)	154 (27.1)	0.798
	Female	5	27.83		90 (36.4)	157 (63.6)		178 (72.1)	69 (27.9)	
Age	≤65 years	3	25.95	0.619	145 (41.1)	208 (58.9)	0.446	259 (73.4)	94 (26.6)	0.695
	>65 years	5	26.00		178 (38.4)	285 (61.6)		334 (72.1)	129 (27.9)	
Sample type	Surgical specimen	5	24.61	<0.001	103(36.4)	180 (63.6)	<0.001	215(76.0)	68 (24.0)	0.002
	Biopsy	10	31.47		108 (33.8)	212 (66.2)		211 (65.9)	109 (34.1)	
	Cytology	0	19.55		112 (52.6)	101 (47.4)		167 (78.4)	46 (21.6)	
Site of sample	Lung—bronchus	5	26.70	0.359	207 (38.3)	334 (61.7)	0.217	387 (71.5)	154 (28.5)	0.577
	Lymph node	2	23.22		54 (47.0)	61 (53.0)		87 (75.7)	28 (24.3)	
	Distant metastasis	5	25.52		62 (38.8)	98 (61.2)		119 (74.4)	41 (25.6)	
Histology	Adenocarcinoma	2	22.16	0.001	209 (44.0)	266 (56.0)	0.006	366 (77.1)	109 (22.9)	0.003
	Squamous carcinoma	10	29.51		59 (31.1)	131 (68.9)		130 (68.4)	60 (31.6)	
	Other NSCLC	10	33.55		55 (36.4)	96 (63.6)		97 (64.2)	54 (35.8)	

N, number; NSCLC, non-small cell lung cancer; TPS, tumor proportion score; PD-L1, programmed cell death ligand 1.

## Data Availability

The data presented in this study are available on request from the corresponding author. The data are not publicly available due to privacy.

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
