# Peer review of "PD-L1 Expression in Non-Small Cell Lung Cancer: Data from a Referral Center in Spain"

_diagnostics, 2021, doi:10.3390/diagnostics11081452_

Round 1

Reviewer 1 Report

The study presents the evaluation of PD-L1 expression by immunohistochemistry in many NSCLC, in different types of tumor samples, and at different anatomical sites.

The PD-L1 immunohistochemistry results are corroborated by several statistical tests and are commented with the literature.

Comments:

  • Is this a retrospective study wherein reports were retrieved from the computer software to read immunohistochemistry results? Would you mind explaining better?
  • The readers can be confused by numbers (Flow chart and text)). The sentence relating to 48 samples (not available) is unclear; excluding these samples from the count is advisable. Also, the reason for the exclusion of the 463 samples from external laboratories is not evident. Then, these are mentioned later in the paper (result-section page. 6, lines 181-183) because they were analyzed anyway. Would you mind explaining better? The Authors should remove this sentence from the results section and move it to the discussion to comment on their findings.
  • In several parts of the paper, there are references to cytology. However, there is no mention of what kind of specimens the Authors are referring to. It would be appropriate to explicit the type of sample per type of anatomical site (bronchial specimens, EBUS, effusions, ..). This can be extended to surgical specimens as well (wedge resection, lobectomy, …). Furthermore, the heading “location” in the Tables is somewhat misleading for lymph node and metastasis (both are metastatic sites). Therefore, it is essential to explain the relationship between sample type and location. Finally, in the discussion, the authors acknowledge that some cytological samples were made on the surgical specimens (page 9, lines 281..); this is not “real life.” If the Authors want to save this part, they should explicit it in the method section and show their results on cytological routine specimens by giving numbers; otherwise, there might be a bias for cytology.
  • Figure 3. f) cell-block from what kind of specimen?
  • The discussion section (Pag. 9, lines 258-265) states that the study cohort comprehends 25% of cytological samples. The Authors should broaden this part of the discussion, and they should try to explain and discuss more accurately why cytology is not performing well. For example, was the PD-L1 evaluation performed by cytopathologists or by the same "expert" pathologist named in the paper?
  • Would you please make a comment about high-grade NSCLC known to show high expression of PD-L1 regardless of histology typing (literature) versus your experience?
  • Would you please explain/discuss why they found higher expression in biopsy specimens versus surgical specimens?

Author Response

We thank the reviewer for their helpful comments and careful review, which helped to improve our manuscript.

REVIEWER 1

The study presents the evaluation of PD-L1 expression by immunohistochemistry in many NSCLC, in different types of tumor samples, and at different anatomical sites.

The PD-L1 immunohistochemistry results are corroborated by several statistical tests and are commented with the literature.

Comment 1

- Is this a retrospective study wherein reports were retrieved from the computer software to read immunohistochemistry results? Would you mind explaining better?

- Reply. Thank you very much for your comment. All PD-L1 tests were performed and analyzed by an optical light microscope at our institution.

The results of samples from our institution are reported in our specific software (Vitropath), while for samples from other institutions, as a referral center in Spain, it is not mandatory to report them in our software, but on a PD-L1 website (platform testing), in which the responsible physician only has access to the results of their patients. Without exception, no patient data from other institutions is provided.

For this study we retrieved the clinicopathological characteristics and PD-L1 results from our institutional software (VitroPath).

Comment 2

- The readers can be confused by numbers (Flow chart and text)). The sentence relating to 48 samples (not available) is unclear; excluding these samples from the count is advisable. Also, the reason for the exclusion of the 463 samples from external laboratories is not evident. Then, these are mentioned later in the paper (result-section page. 6, lines 181-183) because they were analyzed anyway. Would you mind explaining better? The Authors should remove this sentence from the results section and move it to the discussion to comment on their findings.

- Reply. Agreed. We have modified figure 1 and the text as follows:

“A total of 1,307 requests for PD-L1 IHC were received at our center from January 2017 to July 2020 corresponding to 1153 patients with advanced NSCLC. Twenty-eight samples were non-assessable because they had less than 100 viable tumor cells (2.14%).

Of the 1,279 PD-L1 evaluable samples, 463 were from external laboratories and there was no access to all clinicopathological data”.

Additionally, in the results section, we have changed the order of the presentation of the results.

Comment 3

- In several parts of the paper, there are references to cytology. However, there is no mention of what kind of specimens the Authors are referring to. It would be appropriate to explicit the type of sample per type of anatomical site (bronchial specimens, EBUS, effusions, ..). This can be extended to surgical specimens as well (wedge resection, lobectomy, …). Furthermore, the heading “location” in the Tables is somewhat misleading for lymph node and metastasis (both are metastatic sites). Therefore, it is essential to explain the relationship between sample type and location. Finally, in the discussion, the authors acknowledge that some cytological samples were made on the surgical specimens (page 9, lines 281..); this is not “real life.” If the Authors want to save this part, they should explicit it in the method section and show their results on cytological routine specimens by giving numbers; otherwise, there might be a bias for cytology.

- Reply. Most of the biopsies were endoscopic (N = 159; 49.7%), followed by core needle biopsies (N=155; 48.4%) and skin biopsies (N = 6; 1.9%). Regarding surgical specimens, the vast majority of the lung surgical specimens were lobectomies (N=149; 52.7%), but there were also segmentary resections of the lung (N = 48; 17%), pneumonectomies (N = 13; 4.6%) and bilobectomies (N = 3; 1.1%). Other surgical specimens were specific to the site of distant metastasis, including brain resections (N = 33; 11.7); surgical resection of adenopathies (N = 15; 5.3%), pleural resections (N = 8; 2.8%); adrenalectomies (N = 6; 2.1%), nephrectomies, intestinal resections, skin excisions and bone resections (two cases of each; 0.7%). Cytology samples were from endobronchial ultrasound (EBUS)-guided transbronchial needle aspiration (N = 101; 47.4%), tumor needle aspiration (N = 66; 31.0%), pleural effusions (N = 23;10.8%), endoscopic ultrasound-guided needle aspiration (N = 7; 3.3%), bronchoalveolar lavages (N = 7; 3.3%), fine-needle aspiration of superficial adenopathies (N = 6; 2.8%), superficial nodules (N = 2; 0.9%) and pericardial effusions (N = 1; 0.5%).

We have added this information in the text.

- Tables. The term “location” has been replaced by the term “sample site”; and the term “metastasis” by the term “distant metastasis”.

- Of the pairs with surgical specimens and cytology, 102 samples (51 cases) were from the correlation studio, and have been deleted from the comparison between surgical specimens and cytology samples. The new total of paired samples in this category is 19 (38 samples). Therefore, with the new results, we have changed figure 4b and the text in the results section as follows:

“Among the cases included in our study, 191 samples corresponded to 88 unique patients. The samples were paired according to the type of sample (34 cases of biopsy and cytology, 11 cases of biopsy and surgical resection, and 19 cases of cytology and surgical specimen), and site of the sample (six cases of lymph node metastases and distant metastases, 21 cases of lymph node metastases and primary tumor, and 17 cases of primary tumor and distant metastases), resulting in 108 pairs of samples for comparison. Figure 3 shows representative immunohistochemistry images of these comparative groups.

Most of the pairs of samples analyzed were between different types of samples (N = 64; 59.3%). The three comparisons made between the type of sample had a good Pearson correlation coefficient (p-value <0.05), which measured PD-L1 TPS as a continuous variable (Figure 4a-c). However, the agreement calculated with Cohen’s kappa coefficient for the TPS 1% and the TPS 50% cut-off points was much lower. Cytology and surgical specimen comparison had a moderate agreement for the PD-L1 TPS 1% cut-off point (k=0.451) and a substantial agreement for the PD-L1 TPS 50% cut-off (k=0.617). Comparisons of biopsy and cytology had moderate agreement (k=0.436 for TPS 1% cut-off and k=0.544 for TPS 50% cut-off). Biopsy and surgical specimen had slight agreement for TPS 1% cut-off (k=0.1) and moderate agreement for TPS 50% cut-off (k=0.421).”

Comment 4

- Figure 3. f) cell-block from what kind of specimen?

- Reply. Cell blocks were prepared using HistoGel with normal saline, and cells were obtained by fine-needle aspiration, pleural and pericardial effusions or bronchoalveolar lavages. It has been added in Line 80-81.

Comment 5

- The discussion section (Pag. 9, lines 258-265) states that the study cohort comprehends 25% of cytological samples. The Authors should broaden this part of the discussion, and they should try to explain and discuss more accurately why cytology is not performing well. For example, was the PD-L1 evaluation performed by cytopathologists or by the same "expert" pathologist named in the paper.

- Reply. PD-L1 evaluation was made by pathology specialists trained on PD-L1evaluation. Among them, they were two experts with lung cancer cytology: a cytotechnician (N.V) and a cytopathologist (D.M.). This information has been added in the text (methods and discussion sections).

The discussion has been modified as follows:

“In fact, in our cohort, cytological samples represent more than 25% of cases. According to our results, cytological samples tend to show less PD-L1 staining in tumor cells, as several authors have previously observed 1, 2. The heterogeneity of PD-L1 expression is a recognized diagnostic challenge, but this finding raises concerns that cytology samples may not be processed well at our institution; may be the procedure harm the tumor cells membrane and cells do not have a completed cell membrane, or the samples from our study have not provided sufficient tumor cells for accurate PD-L1 evaluation, to counteract the tumor heterogeneity. Despite this, because of the concern for undertreatment and PD-L1 TPS interpretation in cytological samples is challenging, these cases were thoroughly examined and evaluated by experts in lung cancer cytology”.

Comment 6

- Would you please make a comment about high-grade NSCLC known to show high expression of PD-L1 regardless of histology typing (literature) versus your experience?

- Reply. Yes. In our study, a higher expression of PD-L1 was found in NSCLC samples with non-adenocarcinoma and non-squamous subtypes. This observation has also been observed and published by other authors. Tsao et al 3  found a higher expression of PD-L1 in poorly differentiated early stage NSCLC samples including large cell, adenosquamous, pleomorphic and sarcomatoid carcinomas. The study by Hong et al 4 observed a higher expression of PD-L1 in squamous cell carcinomas than adenocarcinomas, while the study recently published by Zheng and colleagues 5, found a higher positivity of PD-L1 in squamous cell carcinoma and other histological subtypes of NSCLC compared to adenocarcinoma. Nonetheless, these results seem controversial, as other studies have not found these differences 6, or have reported higher staining in adenocarcinomas 7.

We have added this information in the discussion.

Comment 7

- Would you please explain/discuss why they found higher expression in biopsy specimens versus surgical specimens?

- Reply. Thank you for your comment. We have added this possible explanation in the discussion. The discrepancy could be due to an advanced stage of the biopsy samples compared to surgical specimens, the age of the samples and to intratumor heterogeneity.

“Contradictory findings have been reported regarding PD-L1 expression in biopsy versus surgical specimens. Although Gradecki et al 8have reported a concordance of PD-L1 expression between core biopsy and resection specimens, other authors have found discrepancies between different tissue sampling methods. Gagne et al 6 found a greater PD-L1 expression in cell blocks compared to biopsies and surgical specimens, but supporting our findings, a higher proportion of PD-L1 TPS ≥50% was found in biopsies compared with surgical specimens.

Along these lines, Zheng et al 5 found that PD-L1 expression was higher in biopsies than in surgical resected specimens in a real-world multi-center study in China.

The discrepancy found between PD-L1 expression in biopsy compared to resection specimens could be due to intratumoral heterogeneity, an advanced stage of the biopsy samples compared to the surgical specimens or the age of the samples. Older versus recent specimens may lead to an underestimation of PD-L1 status 9. Although PD-L1 testing is recommended within samples with less than three years old, a diminution in PD-L1 staining has already been observed by Giunchi et al with 1-year-old specimens10”.

References

  1. Martin-Deleon, R.;Teixido, C.;  Lucena, C. M.;  Martinez, D.;  Fontana, A.;  Reyes, R.;  García, M.;  Viñolas, N.;  Vollmer, I.;  Sanchez, M.;  Jares, P.;  Pérez, F. M.;  Vega, N.;  Marin, E.;  Marrades, R. M.;  Agustí, C.; Reguart, N., EBUS-TBNA Cytological Samples for Comprehensive Molecular Testing in Non-Small Cell Lung Cancer. Cancers (Basel) 2021, 13 (9).
  2. Koomen, B. M.;van der Starre-Gaal, J.;  Vonk, J. M.;  von der Thüsen, J. H.;  van der Meij, J. J. C.;  Monkhorst, K.;  Willems, S. M.;  Timens, W.; t Hart, N. A., Formalin fixation for optimal concordance of programmed death-ligand 1 immunostaining between cytologic and histologic specimens from patients with non-small cell lung cancer. Cancer Cytopathol 2021, 129 (4), 304-317.
  3. Tsao, M. S.;Le Teuff, G.;  Shepherd, F. A.;  Landais, C.;  Hainaut, P.;  Filipits, M.;  Pirker, R.;  Le Chevalier, T.;  Graziano, S.;  Kratze, R.;  Soria, J. C.;  Pignon, J. P.;  Seymour, L.; Brambilla, E., PD-L1 protein expression assessed by immunohistochemistry is neither prognostic nor predictive of benefit from adjuvant chemotherapy in resected non-small cell lung cancer. Ann Oncol 2017, 28 (4), 882-889.
  4. Hong, L.;Negrao, M. V.;  Dibaj, S. S.;  Chen, R.;  Reuben, A.;  Bohac, J. M.;  Liu, X.;  Skoulidis, F.;  Gay, C. M.;  Cascone, T.;  Mitchell, K. G.;  Tran, H. T.;  Le, X.;  Byers, L. A.;  Sepesi, B.;  Altan, M.;  Elamin, Y. Y.;  Fossella, F. V.;  Kurie, J. M.;  Lu, C.;  Mott, F. E.;  Tsao, A. S.;  Rinsurongkawong, W.;  Lewis, J.;  Gibbons, D. L.;  Glisson, B. S.;  Blumenschein, G. R.;  Roarty, E. B.;  Futreal, P. A.;  Wistuba, I. I.;  Roth, J. A.;  Swisher, S. G.;  Papadimitrakopoulou, V. A.;  Heymach, J. V.;  Lee, J. J.;  Simon, G. R.; Zhang, J., Programmed Death-Ligand 1 Heterogeneity and Its Impact on Benefit From Immune Checkpoint Inhibitors in NSCLC. J Thorac Oncol 2020, 15 (9), 1449-1459.
  5. Zheng, Q.;Huang, Y.;  Zeng, X.;  Chen, X.;  Shao, S.;  Jin, Y.;  Xue, Q.;  Wang, Y.;  Guo, Y.;  Gu, B.;  Wu, C.; Li, Y., Clinicopathological and molecular characteristics associated with PD-L1 expression in non-small cell lung cancer: a large-scale, multi-center, real-world study in China. J Cancer Res Clin Oncol 2021, 147 (5), 1547-1556.
  6. Gagné, A.;Wang, E.;  Bastien, N.;  Orain, M.;  Desmeules, P.;  Pagé, S.;  Trahan, S.;  Couture, C.;  Joubert, D.; Joubert, P., Impact of Specimen Characteristics on PD-L1 Testing in Non-Small Cell Lung Cancer: Validation of the IASLC PD-L1 Testing Recommendations. J Thorac Oncol 2019, 14 (12), 2062-2070.
  7. Perrotta, F.;Nankivell, M.;  Adizie, J. B.;  Maqsood, U.;  Elshafi, M.;  Jafri, S.;  Lerner, A. D.;  Woolhouse, I.;  Munavvar, M.; Evison, M.;  Booton, R.;  Baldwin, D. R.;  Janes, S. M.;  Kerr, K. M.;  Bianco, A.;  Yarmus, L.; Navani, N., Endobronchial Ultrasound-Guided Transbronchial Needle Aspiration for PD-L1 Testing in Non-small Cell Lung Cancer. Chest 2020, 158 (3), 1230-1239.
  8. Gradecki, S. E.;Grange, J. S.; Stelow, E. B., Concordance of PD-L1 Expression Between Core Biopsy and Resection Specimens of Non-Small Cell Lung Cancer. Am J Surg Pathol 2018, 42 (8), 1090-1094.
  9. Tsao, M. S.;Kerr, K. M.;  Dacic, S.;  Yatabe, Y.; Hirsch, F. R., IASLC atlas of PD-L1 immunohistochemistry testing in lung cancer. Editorial Rx Press: 2017.
  10. Giunchi, F.;Degiovanni, A.;  Daddi, N.;  Trisolini, R.;  Dell'Amore, A.;  Agostinelli, C.;  Ardizzoni, A.; Fiorentino, M., Fading With Time of PD-L1 Immunoreactivity in Non-Small Cells Lung Cancer Tissues: A Methodological Study. Appl Immunohistochem Mol Morphol 2016.

Reviewer 2 Report

Karmele Saez de Gordoa et col. report their experience with PD-L1 expression in non-small cell lung carcinomas from a large retrospective study.

This article is interesting and well written.

I have only three small remarks:

  • Lines 16-17. The number of 1,279 samples is misleading. The figures should reflect the real number of cases of the series after selection: 816 samples (line 129).
  • Line 77. Patients are said to have a diagnosis of advanced or metastatic NSCLC. However, patients’ clinical or pathological stages are not reported. Some cases with surgical samples are probably from resected localized tumors. PDL1 testing in localized tumors is absolutely not a problem. So, if all tumors' advanced or metastatic status cannot be proved, this status must not be specified.
  • Line 159. Tumor Proportion Score (TPS) is said to having been analyzed as a continuous variable. Was it really a continuous variable? The authors should specify how was TPS assessed +++. In practice, TPS is usually rounded (e.g. 0%, 1-5%, 1-10%, 20% …). In practice, when positive cells are rare (<5%) it is also difficult to differentiate isolated positive tumor cells from intra-tumor macrophages. These points could be discussed.

Author Response

REVIEWER 2

We thank the reviewer for the helpful comments on our manuscript, our specific responses to which are as follows:

Comment 1

- Lines 16-17. The number of 1,279 samples is misleading. The figures should reflect the real number of cases of the series after selection: 816 samples (line 129).

- Reply. Agreed. A total of 1,279 samples were analyzed for PD-L1 immunohistochemistry and included in our study. We have modified figure 1 and the text from section 3.1 as follows:

“A total of 1,307 requests for PD-L1 IHC were received at our center from January 2017 to July 2020 corresponding to 1153 patients with advanced NSCLC. Twenty-eight samples were non-assessable because they had less than 100 viable tumor cells (2.14%).

Of the 1,279 PD-L1 evaluable samples, 463 were from external laboratories and there was no access to clinicopathological data. Among the 816 samples analyzed in our center, throughout the study period, an increasing number of PD-L1 test requests was observed: 2017, N = 211; 2018, N= 240; 2019, N= 262 and during the first half of 2020, N = 103”.

Additionally, we have made changes to section 3.2. The order of the presentation of the results has been modified.

Comment 2

- Line 77. Patients are said to have a diagnosis of advanced or metastatic NSCLC. However, patients’ clinical or pathological stages are not reported. Some cases with surgical samples are probably from resected localized tumors. PDL1 testing in localized tumors is absolutely not a problem. So, if all tumors' advanced or metastatic status cannot be proved, this status must not be specified.

- Reply: Thank you very much for the comment. Most patients were tested for PD-L1 status when a diagnosis of advanced or metastatic NSCLC was made. However, we cannot demonstrate this in all patients in our study cohort.  Therefore, we have changed the text accordingly and removed the “advanced or metastatic” word within the text.

Comment 3

- Line 159. Tumor Proportion Score (TPS) is said to having been analyzed as a continuous variable. Was it really a continuous variable? The authors should specify how was TPS assessed +++. In practice, TPS is usually rounded (e.g. 0%, 1-5%, 1-10%, 20% …). In practice, when positive cells are rare (<5%) it is also difficult to differentiate isolated positive tumor cells from intra-tumor macrophages. These points could be discussed.

- Reply: Thank you very much for the comment. We have added these points in the discussion.

In our institution, PD-L1 expression in tumor cells is evaluated according to the guidelines 1, 2, and the report included the proportion of tumor cells with partial or complete membrane staining (TPS-tumor proportion score) 3.This staining is reported as a continuous variable (not rounded, that is, an exact number in between 0 and 100 is given).

Cases with low PD-L1 positivity in tumor cells in FFPE and mainly cytological samples, are thoroughly examined and consulted by another expert pathologist to our center.

The text was modified as follows: “As the expression of PD-L1 is a continuous variable (not rounded, an exact number in between 0 and 100 is given), any scoring around any of the relevant thresholds will inevitably be subject to some interobserver variability. This can be a problematic issue, as treatment decisions are made according the different cut-off points. In our center, the histological evaluation was carried out by pathology specialists trained on PD-L1 evaluation, and in cases with low PD-L1 positivity (TPS <5%), mainly in cytological samples, were thoroughly examined and consulted with another expert pathologist. It should be noted that in this scenario, in samples with a TPS <5%, it is difficult to differentiate the positive tumor cells isolated from the immune cells that infiltrate the tumor”.

References

  1. Tsao, M. S.;Kerr, K. M.;  Dacic, S.;  Yatabe, Y.; Hirsch, F. R., IASLC atlas of PD-L1 immunohistochemistry testing in lung cancer. Editorial Rx Press: 2017.
  2. Lantuejoul, S.;Sound-Tsao, M.;  Cooper, W. A.;  Girard, N.;  Hirsch, F. R.;  Roden, A. C.;  Lopez-Rios, F.;  Jain, D.;  Chou, T. Y.;  Motoi, N.;  Kerr, K. M.;  Yatabe, Y.;  Brambilla, E.;  Longshore, J.;  Papotti, M.;  Sholl, L. M.;  Thunnissen, E.;  Rekhtman, N.;  Borczuk, A.;  Bubendorf, L.;  Minami, Y.;  Beasley, M. B.;  Botling, J.;  Chen, G.;  Chung, J. H.;  Dacic, S.;  Hwang, D.;  Lin, D.;  Moreira, A.; Nicholson, A. G.;  Noguchi, M.;  Pelosi, G.;  Poleri, C.;  Travis, W.;  Yoshida, A.;  Daigneault, J. B.;  Wistuba, I. I.; Mino-Kenudson, M., PD-L1 Testing for Lung Cancer in 2019: Perspective From the IASLC Pathology Committee. J Thorac Oncol 2020, 15 (4), 499-519.
  3. Martin-Deleon, R.;Teixido, C.;  Lucena, C. M.;  Martinez, D.;  Fontana, A.;  Reyes, R.;  García, M.;  Viñolas, N.;  Vollmer, I.;  Sanchez, M.;  Jares, P.;  Pérez, F. M.;  Vega, N.;  Marin, E.;  Marrades, R. M.;  Agustí, C.; Reguart, N., EBUS-TBNA Cytological Samples for Comprehensive Molecular Testing in Non-Small Cell Lung Cancer. Cancers (Basel) 2021, 13 (9).